# Microsporidia as a Potential Threat to the Iberian Lynx (*Lynx pardinus*)

**DOI:** 10.3390/ani12192507

**Published:** 2022-09-20

**Authors:** Fernando Izquierdo, Dolores Ollero, Angela Magnet, Ana L. Galván-Díaz, Sergio Llorens, Lucianna Vaccaro, Carolina Hurtado-Marcos, Elizabeth Valdivieso, Guadalupe Miró, Leticia Hernández, Ana Montoya, Fernando J. Bornay-Llinares, Lucrecia Acosta, Soledad Fenoy, Carmen del Águila

**Affiliations:** 1Facultad de Farmacia, Universidad San Pablo-CEU, CEU Universities, Urbanización Montepríncipe, 28660 Boadilla del Monte, Spain; 2Grupo de Microbiología Ambiental, Escuela de Microbiología, Universidad de Antioquia, Medellín 050010, Colombia; 3Faultad de Veterinaria, Universidad Complutense, 28040 Madrid, Spain; 4Área de Parasitología, Universidad Miguel Hernández de Elche, 03202 Alicante, Spain

**Keywords:** lynx, *Encephalitozoon*, *Enterocytozoon*, seroprevalence, modified trichrome stain, real-time PCR

## Abstract

**Simple Summary:**

The Iberian lynx, which inhabits the Iberian Peninsula, is one of the most endangered felines in the world. Wild Iberian lynx populations have suffered a constant regression over the past century, with a rapid decline of 90% in the last 20 years. Infectious diseases are one of the most critical threats that cause the population decline of these animals, either in the wild or captivity. Different studies have revealed positive seroprevalence against various pathogens, confirming contact and exposure to bacteria, viruses, and parasites. In this sense, searching for pathogens related to the depopulation of the Iberian lynx is vital for conserving and maintaining this threatened species. The present work confirmed the presence of microsporidia, opportunistic intracellular parasites recently related to fungi, in the lynx environment. Also, different species of microsporidia were determined for the first time in the urine, feces, and tissue samples of *Lynx pardinus*. Further studies are needed to establish the impact of microsporidia infection on the survival of the Iberian lynx. These studies would contribute to the endurance and conservation of this feline by implementing new prevention strategies.

**Abstract:**

*Lynx pardinus* is one of the world’s most endangered felines inhabiting the Iberian Peninsula. The present study was performed to identify the presence of microsporidia due to the mortality increase in lynxes. Samples of urine (*n* = 124), feces (*n* = 52), and tissues [spleen (*n* = 13), brain (*n* = 9), liver (*n* = 11), and kidney (*n* = 10)] from 140 lynxes were studied. The determination of microsporidia was evaluated using Weber’s chromotrope stain and Real Time-PCR. Of the lynxes analyzed, stains showed 10.48% and 50% positivity in urine and feces samples, respectively. PCR confirmed that 7.69% and 65.38% belonged to microsporidia species. The imprints of the tissues showed positive results in the spleen (38.46%), brain (22.22%), and liver (27.27%), but negative results in the kidneys. PCR confirmed positive microsporidia results in 61.53%, 55.55%, 45.45%, and 50%, respectively. Seroprevalence against *Encephalitozoon cuniculi* was also studied in 138 serum samples with a positivity of 55.8%. For the first time, the results presented different species of microsporidia in the urine, feces, and tissue samples of *Lynx pardinus*. The high titers of anti-*E. cuniculi* antibodies in lynx sera confirmed the presence of microsporidia in the lynx environment. New studies are needed to establish the impact of microsporidia infection on the survival of the Iberian lynx.

## 1. Introduction

The Iberian lynx (*Lynx pardinus*) is an endemic feline of the Iberian Peninsula and has been listed in Spain as an endangered species since 30 March 1990 [1]. Wild populations of Iberian lynx have suffered a constant regression for the past century, with a vertiginous decline of 90% in the last 20 years [2]. At the beginning of the Conservation Program, it was estimated that less than 200 Iberian lynxes were left on the planet, distributed between two isolated populations, Doñana National Park and Sierra Morena (Spain) [3,4]. There are three significant plans for the conservation of the Iberian lynx, the “Wild cats: Status Survey and Conservation Action Plan” [5], “Estrategia para la Conservación del Lince Ibérico (*Lynx pardinus*) en España” [6], and “Action Plan for the Iberian Lynx in Europe (*Lynx pardinus*)” [7]. These three plans for conserving the Iberian lynx include five autonomous communities in Spain (Andalucia, Castilla-La Mancha, Castilla y León, Extremadura, and Madrid) and Portugal [2].

According to the National Strategy for the Conservation of the Iberian lynx, infectious diseases are one of the most critical threats causing the population decline of the Iberian lynx, both in wild and captive animals [2]. Different studies carried out in Andalusia [(Doñana, Huelva Province) and (Sierra Morena, Jaén Province)] have revealed positive seroprevalence against various pathogens, confirming contact and exposure to (i) bacteria such as *Mycobacterium bovis*, *Bartonella hensalae*, *Leptospira interrogans*, *Ehrlichia* spp., *Chlamydophila* spp., and hemotropic mycoplasmas (*Mycoplasma haemofelis*, *Candidatus Mycoplasma haemominutum,* and *Candidatus Mycoplasma turicensis*); (ii) Viruses such as feline coronavirus (FCoV), feline leukemia (FeLV), feline parvovirus (FPV), feline herpesvirus (FHC), feline calicivirus (FCV), canine adenovirus-1, and canine distemper virus (DV); (iii) protozoan parasites such as *Cytauxzoon felis*, *Neospora caninum,* and *Toxoplasma gondii*; and (iv) Nematoda parasites [8,9,10].

Studies on the fecal samples of lynxes in these two geographic areas revealed the presence of cestodes (*Hymenolepis* spp. and *Taenia* spp.) and nematodes (*Ancylostoma* spp., *Toxocara* spp., *Toxascaris leonina,* and *Capillaria* spp.) [11]. Moreover, in Extremadura, similar studies have shown the presence of nematodes (*Ancylostomatidae*, *Toxocara cati*, *Toxascaris leonina,* and *Trichuris* spp.) and protozoa (*Cystoisospora* spp.) [12]. Both studies recommend the implementation of broader epidemiological investigations and new surveillance programs for these parasitic diseases to collaborate in the conservation of the Iberian lynx [11,12].

Nevertheless, searching for other pathogens related to the depopulation of the Iberian lynx is vital for conserving and maintaining this threatened animal species. It is essential to highlight that the Iberian lynx is a predator, with rabbits (*Oryctolagus cuniculus*) constituting 85–100% of its diet, regardless of temporal and/or geographical factors [2].

The rabbit is a lagomorph, the natural host of the microsporidia *Encephalitozoon cuniculi.* However, other species, such as *Enterocytozoon bieneusi* and *Encephalitozoon intestinalis,* have also been described in Spain [13,14,15].

Microsporidia are intracellular opportunistic parasites recently related to fungi that, depending on the genus and species, can present a clear zoonotic role capable of infecting different mammals, including humans [16]. They have a wide geographic distribution and most frequently parasitize rabbits, rats, mice, dogs, hamsters, guinea pigs, cats, horses, elephants, primates, and carnivores [17]. A study conducted in Spain on domestic animals (dogs and cats), farm animals (rabbits, pigs, and ostriches), and wild animals (foxes) revealed the presence of species of microsporidia pathogenic to humans. This study corroborates their role as agents of environmental contamination and transmission of microsporidiosis [14].

Considering the high prevalence of *Encephalitozoon cuniculi* in rabbits, the diet of the Iberian lynx and the fact that this parasite infects orally, infection by microsporidia should be considered a possible infectious agent for lynxes. This infection course, which comes with a wide variety of symptoms, from intestinal affection to the spread of the parasite to vital organs such as the brain, kidney, spleen, lung, pancreas, eye, myocardium, or muscle, can cause the death of the host [17,18].

Different histopathological studies have revealed numerous cases of membranous glomerulonephritis and lymphoid depletion in a high number of lynxes. However, the etiological agent and its consequences are still unknown [2]. These clinical manifestations could be compatible with microsporidiosis, which encourages and justifies new research on this parasite in these wild animals.

It is essential to call attention to the particular case of pregnant Iberian lynx females. Several studies have demonstrated the vertical transmission of microsporidia in other mammals, mainly carnivores [17]. The congenital infection leads to fetal death, reduction in the number of pups, or lower weight. For the above points, it is of interest to know the degree of involvement of microsporidia as inducers of pathologies in the Iberian lynx. It could contribute to the survival and conservation of this feline by implementing new prevention strategies.

## 2. Materials and Methods

### 2.1. Ethics Statement

The sample collection was performed under the Environmental Council of the Regional Government of Andalusia and the Iberian lynx Ex-Situ Conservation Program [19].

### 2.2. Samples Collection

From December 2009 to January 2013, various biological samples were received from 140 Iberian lynxes [feces (*n* = 52), urine (*n* = 124), and serum (*n* = 138)]. Due to mortality recorded during the study period, tissues collected from necropsy on 13 lynxes that died during the study [spleen (*n* = 13), liver (*n* = 11), kidney (*n* = 10), and brain (*n* = 9)] were analyzed for microsporidia. All samples were obtained and processed following the guidelines and protocols established in “Health Manual of the Iberian lynx” Version 2.1, May, 2014 [19]. The study was conducted in two areas with populations of Iberian lynx, both in Andalusia (South-Western Spain) [(Doñana, Province of Huelva, 37°0′ N, 6°30′ W; 2000 km^2^) and (Sierra Morena, Province of Jaén, 38°13′ N, 4°10′ W; 1125 km^2^)] (Figure 1).

### 2.3. Microsporidian Spores

Microsporidian spores used for control, either Immunofluorescence or Real-Time PCR, were obtained from the culture [*Encephalitozoon intestinalis* (CDC: V297 [20]), *Encephalitozoon cuniculi* (USP-A1 [21]), and *Encephalitozoon hellem* (PV-5-95 [22]). *Enterocytozoon bieneusi* spores were obtained from a fecal sample of an HIV-infected patient.

### 2.4. Indirect Immunofluorescence Antibody Test (IFAT)

An Indirect Immunofluorescence Antibody Test (IFAT) was performed to determine the seroprevalence against *Encephalitozoon cuniculi* in serum samples. Microscope slides were prepared with a 10^7^
*E. cuniculi* spores/mL suspension and coated with 20 µL per well. Samples were tested with double serial dilutions from 1/50 to 1/6400, and an anti-Cat IgG (whole molecule)—FITC conjugated antibody was used (Ref. F4262, Merck). The samples were examined under a fluorescence microscope with a magnification of 1000×. The positivity cut-off was established with a title of 1:100 [23].

### 2.5. Modified Trichrome Stain

Weber’s chromotrope stain (Modified Trichrome) was used to detect microsporidia spores from urine, feces, and tissue imprints [24]. Before staining, urine samples were washed with PBS 1X and centrifuged at 2500 rpm for 15 min. The supernatant was removed, and the pellet was resuspended with sterile PBS 1X. Thin smears from fecal samples and urine sediment were analyzed at a magnification of 1000× [14].

### 2.6. Genomic DNA Extraction and Real-Time PCR

The urine and feces samples were washed with PBS-EDTA and centrifuged at 2500 rpm for 15 min to remove inhibitors. Genomic DNA was extracted from microsporidian spores (positive controls), stools, and urine samples, according to the protocol previously described by da Silva et al. [25], and from the tissues, as previously described by Andreu-Ballester et al. [26]. All extracted DNA was stored at −80 °C until PCR analysis.

A SYBR Green Real-Time PCR described by Polley et al. [27] and modified by Andreu-Ballester et al. [26] was carried out. This PCR partially amplifies the SSU-rRNA allowing it to distinguish microsporidian species by Tm [*E. hellem*/*E. intestinalis* (82.85–83.9 °C), *E. cuniculi* (84.45 ± 0.4 °C), and *E. bieneusi* (82.35 ± 0.4 °C)]. Positive and negative controls were included in all assays.

## 3. Results

### 3.1. Detection of Anti-Encephalitozoon cuniculi Antibodies in Lynxes Sera by IFAT

Of 138 lynx sera studied by IFAT, seropositivity was obtained in 77 (56%). Serum titer values (ST) showed significant variability ranging from 100 to 6400, and 50 of them (64%) showed titers ≥ 400. Eighteen lynxes showed a 100 ST (13%), and 28 an ST ≥ 1600 (20.28%). From these, 12 (9%) reached 1600 ST, 6 (4%) a 3200 ST, and 10 (7%) a 6400 ST (Figure 2 and Figure 3).

### 3.2. Detection of Microsporidia Spores by Modified Trichrome Stain

One hundred and twenty-four urine samples and 52 feces samples were obtained from the 140 lynxes studied. In total, 10.48% (13/124) of urine samples (13 lynxes) were positive by microscopy, while in feces, the percentage rose to 50% (26/52, 26 lynxes) (Figure 4). Imprint results from tissues of the 13 dead lynxes are shown in Table 1. During the first year of the study, ten lynxes died, and the other three in the second year.

### 3.3. Detection of Microsporidian Species by Real-Time PCR

Briefly, only one urine sample was confirmed by PCR as *Encephalitozoon cuniculi* (1/13, 7.69%); in feces, three species of the genus *Encephalitozoon* and *Enterocytozoon* were identified (17/26, 65.38%). Results are shown in Table 2. *E. bieneusi* was the most prevalent (13/17, 76.5%), co-infecting with *Encephalitozoon* species (7/17, 41.17%). *E. cuniculi* was observed alone in feces from two lynxes (2/17, 11.76%), the same as *E. hellem/E. intestinalis*.

In tissues from dead lynxes, only *Encephalitozoon* was observed. The spleen was the organ with the highest positivity (8/13, 61.53%). In brain tissue, *E.cuniculi* was detected in five lynxes (5/9, 55.55%) with co-infection (*E. hellem/E. intestinalis)* in one of them. Kidneys were also positive for microsporidia (5/10, 50%); *E. cuniculi* was identified in three of them. Five livers were also positive (5/11, 45.45%), with *E. cuniculi* detected in two of them.

## 4. Discussion

The Iberian lynx is the most endangered feline on the Iberian Peninsula. Spain and Portugal have developed different plans and programs, with the support of European and International Organizations, to promote the necessary recovery strategies for the Iberian lynx [“Wild cats: Status Survey and Conservation Action Plan” [5], and “Action Plan for the Iberian Lynx in Europe (*Lynx pardinus*) [7]]. Different factors endanger the Iberian lynx. Among them, the geographical elements, alterations, or destruction of the peninsular Mediterranean mountains due to the advance of livestock, agriculture, urbanism, or mining exploitation stand out [2].

Other factors include the decrease in the rabbit population (which accounts for 85–100% of the diet) or the potential inbreeding of the lynx with less genetic variability that makes it more susceptible to infections. With regard to the latter, factors such as diseases would aggravate the recovery of this feline [2]. Health control is one of the main objectives in the different plans and strategies for recovering the Iberian lynx. Such health control could help decrease the transmission of potentially pathogenic infectious agents transmitted by other animals (carnivores, ungulates, and domestics).

Different studies have described positive seroprevalence against various infectious agents, confirming their exposure to lynxes. Moreover, many pathogens have been detected in biological samples of lynxes, including viruses, bacteria, protozoans, and helminths [8,9,10,11,12,28]. Likely, parasites are the ones less studied. However, it is necessary to highlight their relevance for their involvement in the development of diseases in the populations of the Iberian lynx. Studies had previously focused on entero-parasites, especially cestodes and nematodes, so the need for knowledge about other potential pathogens arose.

The present work studied and monitored 140 lynxes by the analysis of a large number of biological samples [serum (*n* = 138), urine (*n* = 124), feces (*n* = 52), and tissues (*n* = 43)] using various techniques (IFAT, staining, and molecular biology methods). The results obtained are of great interest and relevance, opening a new research field in the preservation of the Iberian lynx.

Antibodies against *Encephalitozoon cuniculi* were studied, as it is a microsporidian species widely described in many animal hosts such as other felines and rabbits [17]. Of the 138 serums studied, 77 lynxes (55.8%, titers ≥ 1:100) were seropositive, some of them (28) reaching high titers, ST ≥ 1600 (20.28%) [23]. From them, 12 (9%) attained 1600 ST, 6 (4%) a 3200 ST and 10 (7%) a 6400 ST. These data confirm the high degree of exposure of the lynx to this parasite, as more than half of them have a positive reaction against the spores of *E. cuniculi*. This positivity could be due to environmental factors, such as spores in water, air, and soil, or the lynx’s diet of rabbits. IgG antibodies against *E. cuniculi* would also reveal a continuous and/or repetitive exposure to the parasite, leading to a secondary (adaptive) immune response.

Considering that lynxes are exposed to microsporidia, the presence of the parasite was studied in different biological samples. For the first time, the present work describes various species of microsporidia in urine, feces, and tissue (spleen, brain, liver, and kidney) samples of lynxes. *Encephalitozoon* species were found, and it is essential to note their exceptional capacity for dissemination to the vital organs. *Enterocytozoon bieneusi* was also detected. Both genera have already been described in other wild animals in the same geographical area as the lynxes [14].

Structures which, by color and size, are compatible with microsporidia spores were detected in urine-stained slides. These characteristics, shape (spherical or oval spores), and the presence of structures such as the posterior vacuole, the filament or polar tubule, and the characteristic bright pink color of the spores, permitted their detection. The molecular technique (Real-Time PCR) was used as a complementary technique to confirm these results and characterize the species involved.

Spores were identified by staining in 13 of the 124 urine samples (10.48%), and one was characterized as *E. cuniculi* (1/13, 7.69%) by Real-Time PCR. Histopathological studies of dead lynxes have previously described membranous glomerulonephritis with an unknown etiological agent (National Strategy for the Conservation of the Iberian Lynx, 2009). With the results obtained from urine samples, microsporidia detection should be included in the autopsies on lynxes as a new etiological agent of kidney affection.

Fecal-stained slides showed 50% positivity (26/52). Of the 26 samples, 17 (65.38%) were confirmed by PCR and characterized as *E. bieneusi* or one of the three species of *Encephalitozoon*. Co-infection of two species was also detected (7/17, 41.17%) as previously described in other animals [17,29]. While co-infection in the kidney is rarely seen, enteric co-infections are widely described in other animals [14].

Studies using complementary techniques, staining and Real-Time PCR, allow for a broader identification of the different biological samples according to their nature and conservation conditions. The presence of inhibitors in biological samples such as fecal matter is described in the literature [14,30]. It should be noted that sample transportation to the laboratory took, in some cases, a few weeks. This time lapse could have affected the results obtained when using the molecular technique.

Another factor to consider when performing microsporidia PCR is the spontaneous extrusion of the spores with the loss of the genomic material. In this case, microsporidia were only identified by microscopy but not with the molecular technique. Moreover, some of the positive samples observed with the staining techniques may correspond to other microsporidian species not included in the PCR protocol [14].

These facts could explain the different results obtained between staining and molecular techniques, highlighting the importance of the complementary use of both methods.

Microsporidia have been mainly detected in the intestinal tract, depending on the species, and have shown different degrees of dissemination to other tissues or organs. *Encephalitozoon* is more frequently found in tissue samples or tissues due to its ability to spread [(preferential *E. cuniculi* in brain and kidney; *E. intestinalis* in the hepatobiliary tree, kidney, and bronchial epithelium; and *E. hellem* in corneal epithelium, lung, and kidney)]. *Enterocytozoon bieneusi,* mainly in the intestinal environment, has also been described in other organs, such as the lung [17,29,31]. A further factor that would facilitate the dissemination of the species of these two genera would be their ability to infect, multiply, and migrate to the organs inside macrophages [32].

The imprints from ten studied kidneys have not produced any positive results by staining, while spores were detected in the spleen, brain, and liver. However, molecular techniques identified microsporidia in all tissue samples analyzed.

An interesting result would be the co-infection detected in the brain in a lynx with *E. cuniculi* and *E. hellem*/*intestinalis* confirmed by Real-Time PCR. With regard to *Enterocytozoon bieneusi* detection in feces by Real-Time PCR, this reinforces an intestinal localization. However, its dissemination to kidney cannot be excluded since spores compatible with *E. bieneusi* have been detected in urine by staining but not by molecular methods, probably because of the aforementioned facts. However, the high presence of *Encephalitozoon* species in vital organs such as the brain (55.55%), spleen (61.53%), kidney (50%), and liver (45.45%) suggests a potential health risk for lynx populations.

## 5. Conclusions

For the first time, this work confirms the presence of microsporidia in different biological samples in the Iberian lynx. Its presence could be a potential threat related to the population decline of the Iberian lynx, making it necessary to evaluate the current prevention measures and implement new strategies for their conservation. These strategies should be aimed at greater sanitary control of the rabbit population in areas with lynx in captivity, mainly with offspring, pregnant females, or lynxes with specific physiological and/or immunological characteristics [33,34,35].

Finally, these findings should help the protection plans of the lynx as they have shown that microsporidia could play a role in the depopulation of the lynx. Routine microsporidia detection protocols and prophylactic measures should be considered to protect lynx populations, especially when feeding lynx in captivity.

## Figures and Tables

**Figure 1 animals-12-02507-f001:**
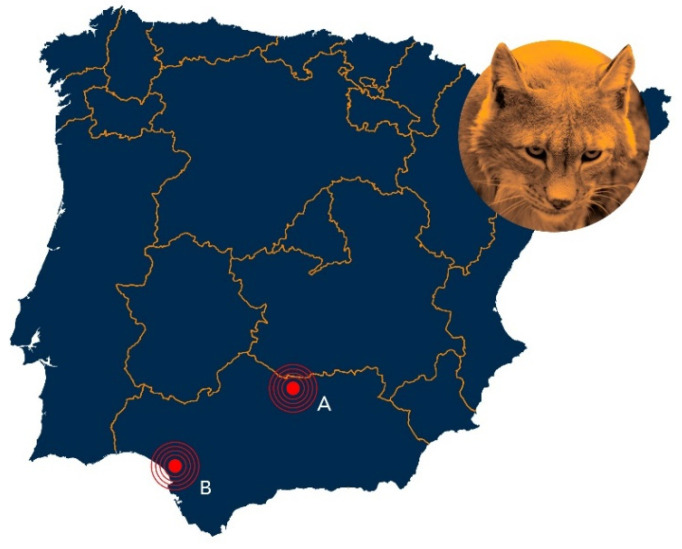
Map of Iberian Peninsula (Portugal and Spain). Andalusia, Southern Spain, shows Sierra Morena (**A**) and Doñana (**B**) study areas.

**Figure 2 animals-12-02507-f002:**
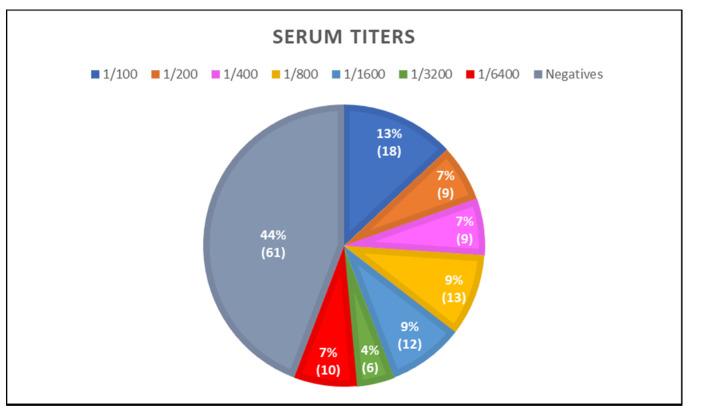
Distribution of anti-*E. cuniculi* serum titers by IFAT.

**Figure 3 animals-12-02507-f003:**
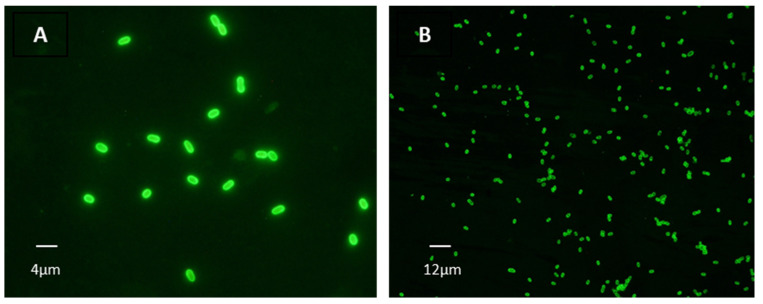
Positive reactivity of lynx sera against *E. cuniculi* spores by IFAT. (**A**): 1:100 titer (magnification 1000×). (**B**): 1:800 (magnification 400×).

**Figure 4 animals-12-02507-f004:**
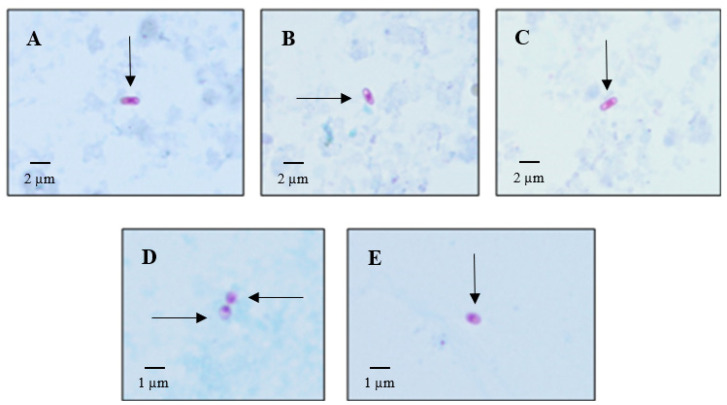
Microsporidia spores stained with Modified Trichrome in urine sediment. (**A**–**C**): spores compatible with the genus *Encephalitozoon* (size: 2–3 µm, magnification: 1000×). (**D**,**E**): spores consistent with the species *Enterocytozoon bieneusi* (length: 1–1.2 µm, magnification: 1000×).

**Table 1 animals-12-02507-t001:** Microsporidia analysis of samples from 13 dead lynxes in urinary sediment, feces, and tissues by staining and Real-Time PCR.

LYNX [Sex/Age] (*n* = 13)	SERUM TITER (*n* = 12)	URINE (*n* = 13)		FECES (*n* = 12)		SPLEEN (*n* = 13)		BRAIN (*n* = 9)		LIVER (*n* = 11)		KIDNEY (*n* = 10)	
		MT-STAIN	RT-PCR	MT-STAIN	RT-PCR	MT-STAIN	RT-PCR	MT-STAIN	RT-PCR	MT-STAIN	RT-PCR	MT-STAIN	RT-PCR
						IMPT	TISSUE	IMPT	TISSUE	IMPT	TISSUE	IMPT	TISSUE
L003 [Male, 10y]	200	−	−	+	*Ecu*	−	−	−	−	NA	NA	−	−
L006 [Male, 5y]	800	−	−	+	−	+	*Ecu*	+	−	+	−	−	−
L007 [Male, 5y]	−	−	−	+	*Ebi*	+	* Ehe/Ei *	−	*Ecu*	+	*Ecu*	−	*Ecu*
L011 [Male, 11y]	100	−	−	−	−	+	*Ecu*	−	*Ecu*	−	* Ehe/Ei *	−	*Ecu*
L031 [Male, 6y]	400	−	−	−	−	−	−	NA	NA	+	−	−	−
L034 [Male, 2y]	800	−	−	NA	NA	−	*Ecu*	+	*Ecu*	−	*Ecu*	−	−
L039 [Male, 2y]	800	+	−	−	−	−	*Ecu*	−	*Ecu*	−	−	−	*Ecu*
L045 [Female, 2y]	3200	-	-	+	*Ebi*	+	-	NA	NA	NA	NA	NA	NA
L047 [Male, 1y]	400	−	−	+	−	+	* Ehe/Ei *	NA	NA	−	−	NA	NA
L091 [Male, 1y]	200	−	−	−	−	−	−	−	−	−	* Ehe/Ei *	NA	NA
L097 [Male, 5y]	−	−	−	−	−	−	* Ehe/Ei *	NA	NA	−	−	−	* Ehe/Ei *
L135 [Female, 9y]	NA	−	−	−	−	−	−	−	−	−	* Ehe/Ei *	−	* Ehe/Ei *
L136 [Female, 20y]	−	−	−	−	−	−	*Ecu*	−	* Ecu * & *Ehe*/*Ei*	−	−	−	−
Positives (%)	9 (75)	1 (7.69)	0 (0)	5 (41.66)	3 (25)	5 (38.46)	8 (61.53)	2 (22.22)	5 (55.55)	3 (27.27)	5 (45.45)	0 (0)	5 (50)

MT-Stain: Modified Trichrome Stain; PCR: Real-Time PCR; IMPT: imprint; y: year(s); NA: sample not available; (+): positive; (−): negative; *Ecu*: *Encephalitozoon cuniculi*; *Ehe*: *Encephalitozoon hellem*; *Ebi*: *Enterocytozoon bieneusi*; *Ei*: *Encephalitozoon intestinalis*.

**Table 2 animals-12-02507-t002:** Microsporidia species in lynx feces samples identified by Real-Time PCR.

LYNX [Sex, Age] (*n* = 17)	Microsporidia Species
L001 [Female, 5y]	*E. hellem/intestinalis*
L003 [Male, 10y]	*E. cuniculi*
L007 [Male, 5y]	*E. bieneusi*
L009 [Female, 5y]	*E. bieneusi*
L013 [Female, 5y]	*E. bieneusi*
L037 [Male, 3y]	*E.**hellem/intestinalis* & *E. bieneusi*
L045 [Female, 2y]	*E. bieneusi*
L046 [Male, 2y]	*E. hellem*/intestinalis
L074 [Male, 1y]	*E.**hellem/intestinalis* & *E. bieneusi*
L077 [Female, 1y]	*E. bieneusi*
L083 [Female, 2y]	*E.**hellem/intestinalis* & *E. bieneusi*
L084 [Male, 1y]	*E. hellem/intestinalis* & *E.**bieneusi*
L086 [Male, 8y]	*E. bieneusi*
L110 [Female, 1y]	*E. cuniculi*
L122 [Female, 1y]	*E.**hellem/intestinalis* & *E. bieneusi*
L123 [Male, 1y]	*E.**hellem/intestinalis* & *E. bieneusi*
L124 [Male, 1y]	*E.**hellem/intestinalis* & *E. bieneusi*

y: year(s).

## Data Availability

Not applicable.

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
