# Peer review of "Microsporidia as a Potential Threat to the Iberian Lynx (Lynx pardinus)"

_animals, 2022, doi:10.3390/ani12192507_

Round 1

Reviewer 1 Report

This is a novel work that includes species of major public health interest. The ONE HEALTH concept is reinforced with work of these characteristics where microsporidian species such as Encephalitozoon cuniculi, which is highly prevalent in pets, can put human health at risk as it can be considered a zoonosis.

The article is very well detailed, with techniques applicable in any specialised laboratory, not forgetting confirmatory molecular biology techniques. The epidemiology in the species studied provides novel data within the animal environment and leaves the door open for studies in other animal species. The carnivorous diet of the lynx in the wild and the high prevalence of E. cuniculi in rabbits highlight the importance of this parasitosis in our immediate environment.

In view of the work, congratulations to the authors.

Author Response

The authors appreciate the Reviewer's comments.

Reviewer 2 Report

This is an interesting study documenting the presence of microsporidia (Encephalitozoon) in endangered Iberian lynx and suggest a link to their diet of rabbits. Editing for scientific communication in English is recommended as some use of certain terms and phrases should be clarified (e.g. "in the lynx environment"; it is unclear what this means. As "biopsy" implies a tissue sample taken from a live animal, the authors should change the wording to reflect tissues collected from post-mortem examinations, which upon reading the methods and results, appears to be the case. The authors present the findings as evidence of a "new threat" to the lynx, but no histopathology is included to indicate that the detection might have been associated with disease. As much of host-parasite dynamics are not completely understood in an ecological setting, negative impacts on hosts are not always clear and it does not appear that the authors demonstrate negative impacts on the host in this publication. Please clarify what is being presented in the serological results (i.e. is there evidence of serological exposure in the absence of disease here or another cause of death?). Remove Figure 1 as it is unnecessary. 

In addition to comments above, I would highly recommend that the authors work with a professional scientific translator for some re-phrasing and clarification of certain terms (as opposed to just sending it to an editor) as there may be nuances that should be clarified.

Author Response

We want to thank the Reviewer for the suggestions and advice to improve the manuscript. We have addressed each request very carefully and corrected our manuscript accordingly. We believe that the Reviewer's recommendations have significantly improved the manuscript.

1) "Editing for scientific communication in English is recommended as some use of certain terms and phrases should be clarified (e.g. "in the lynx environment"; it is unclear what this means".

A native editor has carefully revised the article.

2) "As biopsy implies a tissue sample taken from a live animal, the authors should change the wording to reflect tissues collected from post-mortem examinations, which upon reading the methods and results, appears to be the case.

The authors agree with the Reviewer and appreciate the clarification. In the M&M section, "biopsy" has been changed to: "tissues collected from the necropsy". From there on, we refer to biopsy in the text as tissue samples.

3) The authors present the findings as evidence of a "new threat" to the lynx, but no histopathology is included to indicate that the detection might have been associated with disease. As much of host-parasite dynamics are not completely understood in an ecological setting, negative impacts on hosts are not always clear and it does not appear that the authors demonstrate negative impacts on the host in this publication".

The authors appreciate the comment and agree with the Reviewer. The article presents, to date, the first study of the presence of microsporidia in the Iberian lynx. This pilot study searches for new parasites related to the Iberian lynx that may endanger the population of these felines.

Stating that it is a "new threat" may be rather adventurous, so we have modified the conclusion and it now reads, "Its presence could be a potential threat related to the Iberian lynx's population decline…".

With this statement, we want to show the potential pathogenicity of microsporidia as it has been found in different types of samples. Nevertheless, as suggested by the Reviewer, histopathological studies would or would not confirm microsporidia pathogenicity and allow confirmation of the cause or causes of the death of lynxes.

Due to the difficulty and complexity in obtaining and accessing the samples and the size of the tissue sample received, it only allowed us to perform the imprints and molecular biology techniques. The authors are aware that the mere presence or detection of microsporidia in such tissues is not indicative that it is the causative agent of lynx death and have modified the conclusions accordingly.

4) Please clarify what is being presented in the serological results (i.e. is there evidence of serological exposure in the absence of disease here or another cause of death?).

The authors understand the Reviewer's doubt. The serological study seeks to understand the degree of exposure of the lynx to the parasites. The presence of IgG in the serum samples confirms the continuous exposure to E. cuniculi suggesting the free circulation of microsporidium in the lynx ecosystem. Nor can we ignore the exposure derived from the lynx's diet due to the intake of rabbits, one of the usual hosts of E. cuniculi. In both situations, systemic infection could occur as we have found spores in different organs. Nevertheless, the consequences of lynx health could not be assessed only with the serological test alone.

Therefore, with the serological results obtained, we can confirm the exposure to microsporidia that correlates with the presence of spores found in other biological samples.

5) Remove Figure 1 as it is unnecessary.

The authors respect the Reviewer's suggestion, but we believe that the figure should be maintained since it could be useful for other researchers and readers to know the geographical location of the different populations of lynx studied.

6) In addition to comments above, I would highly recommend that the authors work with a professional scientific translator for some re-phrasing and clarification of certain terms (as opposed to just sending it to an editor) as there may be nuances that should be clarified.

A native editor has thoroughly revised the article.

Reviewer 3 Report

Dear Authors,

thanks for submitting this manuscript. It is overall well presented. I have just small remarks on the manuscript.

Key words: the key words are partially redundant with the title. Key words should complement the title not repeat it.

Small remarks:

In general, it would be an interesting information to add sex and age of the sampled lynxes, e.g. in table 1 and 2.

Ln 226 means probably “dead”

Thanks and all the best

Author Response

The authors appreciate the Reviewer's suggestions.

1) "Key words: the key words are partially redundant with the title. Key words should complement the title not repeat it".

They have been corrected accordingly.

2) Small remarks:

2.1) "In general, it would be an interesting information to add sex and age of the sampled lynxes, e.g. in table 1 and 2".

The authors have included the sex and age of the lynxes in Tables 1 and 2.

2.2) Ln 226 means probably "dead".

It has been corrected accordingly.

Reviewer 4 Report

The research is interesting for the animal species considered, the Lynx, where there are few epidemiological data on diseases.

The manuscript is well written and structured.

The material and methods are clearly described and allow to achieve the results present in the scope of the research.

The chapter of conclusions is missing

Author Response

The authors appreciate the Reviewer's suggestions.

1) "The chapter of conclusions is missing".

It has been corrected accordingly.

Reviewer 5 Report

The Iberian lynx is one of the most endangered felines in the world that inhabits the Iberian Peninsula. This study was performed to identify the presence of microsporidia due to the mortality increase of lynxes. The manuscript is scientifically sound and the findings are relatively well explained. It is sufficiently to publication in the journal. Therefore, I recommend its acceptance for publication after minor reversion.

Specific comments:

1. Introduction. Please rewritten the presence of nematodes, cestodes and protozoan to make them in the same paragraph.

2. Line 56-65. Please check the use of brackets.

3. Line 175. Please check “an”.

Author Response

The authors appreciate the Reviewer's suggestions.

1) "Introduction. Please rewritten the presence of nematodes, cestodes and protozoan to make them in the same paragraph".

The authors understand the Reviewer's suggestion, but we have decided to keep the paragraphs as they were originally written. Paragraphs were organized according to the sample type because the discussion follows the same order. We would prefer to keep the structure.

The first paragraph describes serological studies (seroprevalence) against different pathogens (viruses, bacteria, protozoa, and nematodes). The second paragraph describes the presence and detection of helminths (cestodes and nematodes) and protozoa in fecal samples from different geographical areas of Spain (Andalusia and Extremadura).

2) Line 56-65. Please check the use of brackets.

It has been corrected accordingly..

3) Line 175. Please check "an".

It has been corrected accordingly

Round 2

Reviewer 2 Report

Thank you for making the requested adjustments. It is still unclear to the readers how microsporidia is a threat to the lynx, but certainly there is evidence that it is detected. I still think the title should be adjusted to better represent the paper in terms of "Confirmation of the presence and exposure to microsporidia in the Iberian lynx". It remains unclear exactly what negative impacts the authors claim to be responsible for population declines in lynx based upon this study alone. I would suggest further rewording to the effect of further investigation into potential negative impacts on individual animals and thus the population may be warranted prior to suggesting management measures. 

Author Response

We want to thank the Reviewer for the suggestions and advice to improve the manuscript. We have addressed each request very carefully and corrected our manuscript accordingly.

  • It is still unclear to the readers how microsporidia is a threat to the lynx, but

certainly there is evidence that it is detected. I still think the title should be adjusted to better represent the paper in terms of "Confirmation of the presence and exposure to microsporidia in the Iberian lynx".

As we have reliably detected that microsporidia multiply inside lynxes and do so in vital organs, including the brain, and since these parasites are recognized as pathogens in different animals, rabbits, and carnivores (please see lines: 91-96), we postulate that microsporidia could be a potential threat to lynxes.

Therefore, new studies should be carried out aimed at clarifying this hypothesis. Including Microsporidia in the screening test of lynx pathogens could help better understand the negative impact of the microsporidia on the lynx's health.

Nevertheless, the title was modified to: "Microsporidia as a potential threat to the Iberian lynx (Lynx pardinus)."

Lines 91-96 "Considering the high prevalence of Encephalitozoon cuniculi in rabbits, the diet of the Iberian lynx and the fact that this parasite infects orally, infection by microsporidia should be considered a possible infectious agent for lynxes. This infection course with a wide variety of symptoms, from intestinal affection to the spread of the parasite to vital organs such as the brain, kidney, spleen, lung, pancreas, eye, myocardium, or muscle, can cause the death of the host [17,18]. "

  1. Didier, E.S. Microsporidiosis: an emerging and opportunistic infection in humans and animals. Acta Trop 2005, 94, 61-76, doi:10.1016/j.actatropica.2005.01.010.
  2. Mathis, A.; Weber, R.; Deplazes, P. Zoonotic potential of the microsporidia. Clin Microbiol Rev 2005, 18, 423-445, doi:10.1128/CMR.18.3.423-445.2005.

2) It remains unclear exactly what negative impacts the authors claim to be responsible for population declines in lynx based upon this study alone. I would suggest further rewording to the effect of further investigation into potential negative impacts on individual animals and thus the population may be warranted prior to suggesting management measures.

As aforementioned, microsporidia were found infecting vital organs of the lynx. Therefore, the negative impact could be assumed. Moreover, one cause of lynx's death is nephritic problems. Considering that E. cuniculi is a known pathogen found to cause kidney failure, we can indeed hypothesize that microsporidia could be one of the causes of lynx health issues. Nonetheless, we are not suggesting that microsporidia are the only cause of lynx depopulation. Extinction problems are multifactorial, and the knowledge of all the potential threats to endangered species would improve their conservation. As a consequence of our study, we consider that including Microsporidia in the screening test of lynx pathogens could be a cost-effective measurement in managing the lynx. If a potential pathogen is not included in the routinary test, it would never be diagnosed and, consequently, never treated appropriately.